# Analysis of Conservative Treatment Trends for Lumbar Disc Herniation with Radiculopathy in Korea: A Population-Based Cross-Sectional Study

**DOI:** 10.3390/healthcare11162353

**Published:** 2023-08-21

**Authors:** Sohyun Cho, Yu-Cheol Lim, Eun-Jung Kim, Yeoncheol Park, In-Hyuk Ha, Ye-Seul Lee, Yoon Jae Lee

**Affiliations:** 1Jaseng Korean Medicine Hospital, Seoul 06110, Republic of Korea; kmdjsh05@jaseng.co.kr; 2Jaseng Spine and Joint Research Institute, Jaseng Medical Foundation, Seoul 06110, Republic of Korea; hmh6692@gmail.com (Y.-C.L.); hanihata@gmail.com (I.-H.H.); 3Department of Acupuncture & Moxibustion, College of Korean Medicine, Dongguk University Bundang Oriental Hospital, Seongnam 13601, Republic of Korea; hanijjung@naver.com; 4Department of Acupuncture & Moxibustion, Kyung Hee University at Gangdong, 892 Dongnam-ro, Gangdonggu, Seoul 05278, Republic of Korea; icarus08@hanmail.net

**Keywords:** conservative treatment, radiculopathy, cross-sectional studies, lumbar vertebrae, trend, Korea

## Abstract

This study aimed to analyze the trends in conservative treatment and associated medical costs for lumbar intervertebral disc disorders with radiculopathy in Korea. This population-based cross-sectional study included patients aged ≥ 20 years with at least one “intervertebral disc disorder with radiculopathy” claim (Korean Standard Classification of Diseases (KCD)-7 code: M511) who sought treatment from tertiary, general, or Korean Medicine hospitals or clinics between 2010 and 2019 and whose data were extracted from the Korean Health Insurance Review and Assessment Service National Patients Sample database. Intervention frequency, ratio, and medical costs, including medication, were analyzed. The number of patients with lumbar intervertebral disc disorders and radiculopathy undergoing conservative treatment increased by >30%, and medical costs increased from USD 3,342,907 to USD 5,600,456 during the 10-year period. The non-surgical treatments mainly used were medication and physiotherapy, and the most commonly prescribed medication was non-opioid analgesics. Meanwhile, the number of patients who used nerve plexus and root and ganglion nerve blocks showed the most significant increase. In conclusion, the number of patients with radiculopathy who received nerve blocks, particularly nerve plexus and root and ganglion nerve blocks, and related expenditure increased, implying a gradual shift in medical decisions from systemic pain reduction to specific and targeted pain treatments. Future studies and clinical practice guidelines may require further inspection of real-world practice to advise optimal treatment choices for an effective treatment plan.

## 1. Introduction

Lumbar disc herniation is a displacement or prolapse of the nucleus pulposus (NP) caused by damage to the outer annulus fibrosus surrounding the NP. The herniated NP compresses or irritates the nerve root posteriorly, leading to radiculopathy. Radiculopathy often causes pain, numbness, and tingling in the associated dermatome of the affected nerve roots [1]. Patients with lumbar intervertebral disc disorders (IDDs) with lumbar radiculopathy often suffer from pain in the lower back radiating to the lower limb and paresthesia [2,3,4], and are more likely to have severe functional disability and pain than those with lower back pain alone [5], which ultimately have an impact on their quality of life, daily activities, and productivity [1,2,3,6]. Thus, an increase in the incidence of lumbar IDDs with radiculopathy is not only a problem limited to patients but also a socioeconomic issue, as it can cause productivity loss and affect work performance because patients often need to rest from work and resign early [7,8,9].

The management of symptoms in patients with lumbar IDD and radiculopathy involves the resolution of radiculitis (spinal nerve root inflammation) and improving the lumbar disc herniation. According to clinical practice guidelines, surgery can only be considered as an option when there is no improvement in symptoms after conservative treatment for at least 6 weeks, if not upholding the option of continued non-surgical treatments; furthermore, surgery should not be considered as an option for cases involving the development of neurological symptoms or cauda equina syndrome [1,2,3,5,10]. Several studies have reported the effect of conservative treatment for lumbar IDD [6,11,12,13,14]. Although surgery is effective for faster relief of symptoms in acute settings, no significant difference has been observed between the effectiveness of surgical intervention and conservative treatment over time [1,15,16]. Furthermore, even for surgical candidates with radiculopathy or a sizeable affected area of the herniated disc, the outcomes of conservative management were not inferior to those of surgery [16,17]. In particular, patients with lumbar radiculopathy face postoperative complications, such as a high risk of pain and functional disability, resulting in poor postoperative prognosis [18]. Therefore, for patients who are reluctant to undergo surgical treatment due to fear of surgery, expectations of spontaneous improvement, and lack of certainty regarding the long-term benefits of the surgery [17], conservative treatment is considered to be a suitable primary treatment [1,2,3,17].

Conservative (non-surgical) treatment includes a range of modalities, such as bed rest, physiotherapy, nerve blocks, and medications including opioids, non-opioid analgesics, psychotropic agents, exercise, acupuncture, and manual therapy, whereby the latter two therapies are provided in Korean Medicine clinics in Korea [6,11,19]. However, the effects of each type of intervention remain controversial. Several guidelines and clinical practice recommendations based on lower back pain management, including the 2015 Evidence-Informed Primary Care Management of Low Back Pain, 2016 National Institute for Health and Care Excellence Guideline on Low Back Pain and Sciatica, 2017 American College of Physicians Diagnosis and Treatment of Low Back Pain, and 2018 VA/DoD Clinical Practice Guidelines, have reported inconsistencies or inconclusive evidence or recommendations for some treatments [20]. This may be because only few studies have compared the frequency and effectiveness of non-surgical treatment, which often involves a combination of treatments. Moreover, the trends and current status of non-surgical treatment have not been examined in clinical practice for patients with radiculopathy, which is a relatively severe condition.

This study aimed to examine the trends and status of conservative treatment in clinical practice. Our findings can provide real-world evidence for revising clinical practice guidelines and establishing related healthcare policies.

## 2. Materials and Methods

### 2.1. Data Source

This study analyzed claims data from the Health Insurance Review and Assessment Service (HIRA) and HIRA-National Patient Sample (HIRA-NPS) data from January 2010 to December 2019. Due to universal health coverage in Korea, the National Health Insurance Service (NHIS) covers 98% of the national population in Korea. Among the four types of patient sample data provided by the HIRA, the HIRA-NPS consists of a 2% sample (approximately 1 million people) annually selected via sex-stratified (two classes) and age-stratified (sixteen classes) random sampling from the entire Korean population enrolled in the NHIS program. Raw data underwent de-identification by removing any personally identifiable information, and the information regarding the treatment and prescription of the patient sample based on the NHIS claims was examined in this study [21]. However, as the claim data can only be available in 1-year segments based on the commencement date of treatment for each applicable year, examining the continuous history of claims was not possible.

### 2.2. Study Population

The inclusion criteria were as follows: patients with one or more claims of lumbar IDD with radiculopathy (Korean Standard Classification of Diseases (KCD)-7 code: M511) as the primary diagnosis from 2010 to 2019, and those who visited medical institutions, including tertiary hospitals, general hospitals, hospitals, clinics, and Korean Medicine (KM) hospitals or clinics which provide acupuncture, Tuina or Chuna (manual therapy), and herbal medicine. Only the adult patients aged ≥20 years with no missing data in the study variables were included in the study. Patients with spine-related surgery or hospitalization within the same year were excluded from the final selection of study patients.

### 2.3. Study Outcomes

In this study, the age and sex of patients and the type of payments were included as baseline characteristics for analysis. The age category was divided into six groups in 10-year increments for adults aged 20 years, and the payer type was categorized as NHIS, Medicaid, and others. The conservative treatment for lumbar IDD was categorized into trigger point injection (TPI), physiotherapy, nerve blocks, KMs, and others (“Service category”). KM treatments included acupuncture, moxibustion, cupping, hot/cool pack and infrared therapy, and Chuna manual therapy, which were administered as an integrative treatment in a single medical service; therefore, all types of Korean medical services were comprehensively presented under the category of “KMs”. In the case of nerve blocks, which are the primary mode of conservative treatment, the types of service were categorized into epidural block, peripheral branch block, and nerve collection (plexus, ganglion, and root) blocks, and yearly trends were examined for each type of nerve block. The prescribed medications during inpatient and outpatient care were categorized according to the Anatomical Therapeutic Chemical classification system, and the results are presented in Appendix A. The healthcare expenditure covered by NHIS was analyzed by types of interventions.

### 2.4. Statistical Analysis

Descriptive statistical analysis was performed for the data analysis in this study. For the baseline characteristics of patients’ data, the number of patients by type of medical service, number of patients by prescribed medication category, applicable number of patients, and the percentage of the total number of patients were used. Annual trends in the number of patients in terms of the medical service received and costs incurred were presented using bar and line graphs. All costs in this study were converted to the 2020 average South Korean won to US dollar (KRW/USD) exchange rate and corrected to reflect the consumer price index in the health sector (Appendix A). The statistical software suite SAS (version 9.4, SAS Institute, Cary, NC, USA) was used to calculate results and create graphs.

## 3. Results

The total number of patients with one or more NHIS claims for lumbar IDD with radiculopathy as the primary diagnosis was 274,175 from 2010 to 2019. Among them, the number of patients with category codes for tertiary, general, and KM hospitals and clinics was 272,979, and 173 patients with missing data were excluded. Patients aged <20 years (n = 4156) were also excluded, resulting in 268,650 patients. Finally, after excluding those with NHIS claims records for spine-related surgery or hospitalization, 234,858 patients were selected in the study sample (Figure 1).

### 3.1. Demographic Characteristics of Patients

In this study, patients aged 50–59 years accounted for the highest proportion, followed by those aged 60–69 years, indicating a high proportion of middle-aged and older patients. In addition, there were more female than male patients (58.61% vs. 41.39%). No significant differences were observed in terms of the payment type, but most patients were covered by NHIS (94.12%) (Table 1).

### 3.2. Use of Medical Services

The number of patients who managed with non-surgical intervention steadily increased by more than 30%, from 19,907 in 2010 to 26,441 in 2019 (Figure 2). As the number of patients increased, the annual medical costs gradually increased, except during 2014–2015, leading to a total increase of more than USD 2 million over 10 years (USD 3,342,907 to USD 5,600,456; see Figure 2).

### 3.3. Non-surgical Treatment

Figure 3 shows the yearly trends in the number of patients according to the type of non-surgical treatment from 2010 to 2019. Among the type of non-surgical treatment frequently used in patients with radiculopathy due to lumbar disc herniation for the 10-year period, the most commonly used treatment method was medication, followed by physiotherapy. In particular, the use of medications and nerve blocks has continued to increase over the last 10 years, while the annual number of patients who have undergone physiotherapy has decreased since 2016. The percentage of patients who received physiotherapy was as high as 60.08% (12,103 patients) in 2010, but continuously decreased to 47.46% (12,549 patients) by 2019, indicating that while the gross number did not change, the proportion of patients who received physiotherapy reduced, relative to other noninvasive therapies.

In contrast, the percentage of patients who received nerve blocks showed a continuously increasing trend from 23.29% (4636 patients) in 2010 to 35.36% (9349 patients) in 2019, accounting for the second-highest proportion after physiotherapy among conservative treatment other than prescribed medication. The number of patients who received KM increased from 1185 in 2010 to 1801 in 2019; however, the increase was moderate, from 5.95% to 6.81%. TPI accounted for the lowest proportion among the types of conservative treatment, with lower than 3% of all patients who received noninvasive treatments.

Regarding the annual trends in medical expenses for each medical service category, the annual expenditure on physiotherapy has accounted for more than USD 600,000 over the past decade. The use of nerve blocks has consistently increased from USD 403,233 in 2010 to USD 1,500,957 in 2019, surprisingly representing a nearly threefold increase. In 2012, nerve blocks had the highest medical costs of all forms of conservation treatment. Although the number of patients receiving medications increased during the study period, the medical cost was lower than that of nerve blocks and physiotherapy. In addition, medical costs related to KMs, which did not show any significant change in trend for most of the study period, showed a nearly two-fold increase in 2019 (from USD 156,590 in 2018 to USD 292,438 in 2019) (Appendix A).

### 3.4. Nerve Blocks

Nerve block usage showed the most significant increase among all types of conservative treatment; therefore, data were further analyzed to calculate the number of patients and the 10-year trend for the different types of nerve blocks. The number of patients who were administered with nerve blocks increased for each type: epidural, peripheral, and nerve plexus, root, or ganglion blocks. The 10-year trends differed slightly depending on the type of nerve block used. Spinal nerve plexus, root, and ganglion nerve blocks showed a rapidly increasing trend which was approximately four-fold, from 1239 patients (6.39%) in 2010 to 5442 patients (21.17%) in 2019. In contrast, epidural nerve blocks showed a slight increase to 3769 patients (15.65%) until 2017, followed by a slight decrease, resulting in 3558 patients (13.84%) in 2019. Peripheral branch nerve blocks did not show any specific increase or decrease in the trend, and approximately 7% of the patients received this treatment over the 10-year period (Figure 4; Appendix A).

### 3.5. Medication

Table 2 presents frequently prescribed medications, with M511 as the primary diagnosis. Non-opioid analgesics have been the most commonly prescribed over the last decade, and the number of patients who have been prescribed non-opioid analgesics has steadily increased; this has been followed by opioids, anesthetics, and steroids. Among trends in the use of different medications, the use of anesthetics, steroids, and opioids shows notable patterns. The number of patients who have received anesthetics such as lidocaine has steadily increased, while opioids have not changed over the last decade. Steroids, on the other hand, have shown a gradual increase in the number of patients.

Among the types of prescribed medications, the use of anesthetics has had the most significant increase over the 10-year period. In 2010, opioids accounted for the second-highest proportion of medications prescribed for therapeutic purposes in patients with lumbar IDD and radiculopathy. However, in 2018, patients who used anesthetics outnumbered those who used opioids. In 2018, the number of patients who were prescribed with anesthetics outnumbered those who used opioids, up to a 2.5-fold increase compared to the number of patients in 2010. Regarding the trends in medical costs by type of medication, the cost of non-opioid analgesics was the highest. The expenditure of anesthetics was the lowest in 2010, and showed a gradual increase, resulting in an increase of about 70% in one decade.

## 4. Discussion

Based on 10-year (2010–2019) data from Korea (HIRA-NPS) on the conservative treatment of patients with radiculopathy caused by lumbar disc herniation related to lumbar IDD, we conducted a comprehensive analysis by categorizing conservative treatment into types of medical services and medications used for treatment. With a growing number of patients with lumbar IDD and radiculopathy choosing conservative treatment, the associated medical costs increased by nearly 30%. The most frequently used non-surgical treatment was medication, followed by physiotherapy. Non-opioid analgesics were mainly used for medication. Interestingly, the proportion of patients who received nerve blocks showed a steadily increasing trend. Among the types of nerve blocks, the number of patients who received nerve blocks targeting the collection of nerves (root, plexus, and ganglion) increased significantly, outnumbering the use of epidural nerve blocks since 2015, and the associated costs exhibited a dramatic increase.

The result of this study indicates a change in non-surgical treatment choices over the last decade toward specific interventions. While a high rate of medication prescriptions and physiotherapy is similar to previous studies [9] on patients with lumbar IDD and radiculopathy, the evidence supporting medication is inconsistent despite the high utilization, possibly due to the nonspecific indications of these two treatments. This study shows a dramatic increase in the use of nerve blocks, particularly the perineural approach to the spinal nerve plexus and dorsal root ganglion, compared to the nerve block for radiculopathy.

The epidural injection is the most commonly used treatment for lumbar IDD globally. In the United States, the treatment is performed 9 million times annually [22]. A previous study in Korea in 2009 showed that epidural injection was the most commonly used treatment [23]. While this trend was partially observed in the earlier data of our study, this study also shows a shift in the preferred type of nerve blocks. Even after epidural injections, inflammation in the nerve root is thought to cause pain, which may have been the reason for this change in preference toward a more direct approach to the nerve root and the site of inflammation [9,24]. In addition, the number of patients who received acupuncture and manual therapies in KM clinics showed a mild increase in 2019. This may be due to the inclusion of Chuna or Tuina, a type of manual therapy in KM, under NHIS coverage [25]. In Korea’s dual health insurance system, KM treatments such as acupuncture, cupping, moxibustion, and manual therapy are covered by National Health Insurance.

The continuous increase in anesthetics prescription along with steroids in this study may be related to the increase in nerve blocks. Interestingly, opioids, which are one of the most frequently prescribed medications in patients with lumbar IDD and radiculopathy, were surpassed by anesthetics in 2018. The use of therapeutic agents in nerve blocks have been reported to include both anesthetics and steroids [23,26], supporting the results reported in this study.

Previous studies on the effectiveness of epidural injection of steroids and/or anesthetics compared to conservative treatments such as oral medication show mixed results. Local injection was beneficial for some outcomes such as worse leg pain in short-term pain relief and intermediate-term follow-ups, but not during long-term follow-ups [27,28,29,30]. On the other hand, lumbar plexus blocks have been shown to be more effective than epidural blocks in postoperative anesthesia in terms of surgeon’s satisfaction [31] and the consumption of opiates [32]. However, the lack of sufficient data on the long-term efficacy and safety of lumbar plexus blocks compared to conservative treatment is evident. The increasing number of local injections and related health expenditures imply the need for reassessing clinical guidelines and health policies in line with the current evidence.

Socioeconomic status has been pointed out as a potential influence on the diagnosis, development, and pain outcome of lumbar IDD and radiculopathy in previous studies [33,34,35]. On the other hand, studies showed mixed results regarding the association between overall healthcare utilization and socioeconomic status (SES) [36,37]. This study investigated non-surgical interventions which are covered by NHI, encompassing 98% of the Korean population, and did not include healthcare services not covered by NHI which may depend highly on SES. Future studies are suggested to be conducted with a wider scope of management options of IDD and radiculopathy to examine the potential differences in the utilization rate according to socioeconomic groups.

This study holds significance as it is the first to analyze the current state and 10-year trends related to the use of non-surgical medical treatments for patients with lumbar IDD and radiculopathy based on claims data from the NHIS.

However, this study has some limitations. First, the study only included outpatients and did not account for treatment items not covered by the NHIS, which may limit the applicability of the findings to real-world clinical practice. Second, the NHIS claims data were segmented on an annual basis, which prevented the examination of patients’ surgical history prior to the applicable year. Third, socioeconomic variables were not included in the database, providing limited information on the potential differences in treatments according to socioeconomic status. Finally, patients were selected solely based on NHIS claims diagnosis codes, without considering the severity of lumbar IDD or self-reported pain intensity. Hence, it is crucial to exercise caution while interpreting and generalizing the study findings to broader populations. Further research is required on the comparative effectiveness of therapies such as lumbar plexus blocks compared to epidural injections and conservative treatments prior to making clinical decisions and developing health policies.

## 5. Conclusions

Over the 10-year study period, the number of patients who underwent nerve block treatments, particularly those who used nerve plexus, root, and ganglion nerve blocks, steadily increased. Given the upward trend in the number of patients with lumbar IDD and radiculopathy and the corresponding healthcare costs, the study’s findings call for a need to review the current clinical practice guidelines to align them with the current evidence and formulate relevant healthcare policies in the future. Further research is required regarding the effectiveness of different types of local injections compared to conservative treatments.

## Figures and Tables

**Figure 1 healthcare-11-02353-f001:**
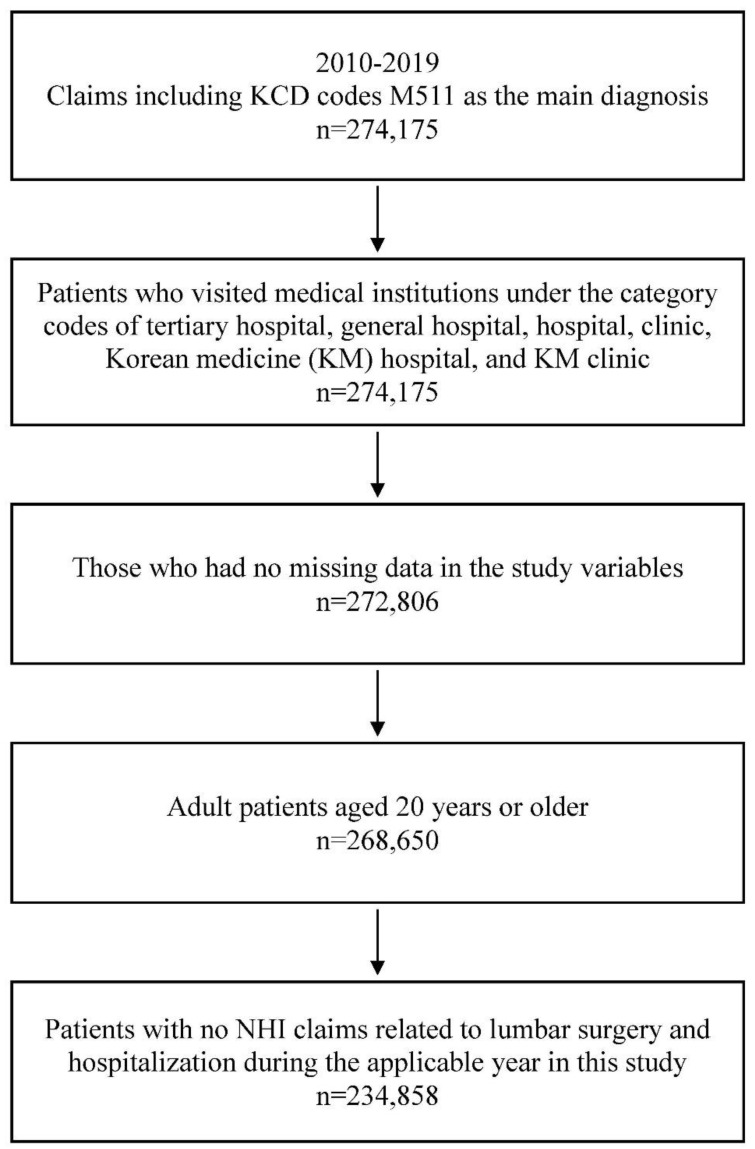
Flow chart; KCD: Korean Standard Classification of Diseases; NHI: National Health Insurance.

**Figure 2 healthcare-11-02353-f002:**
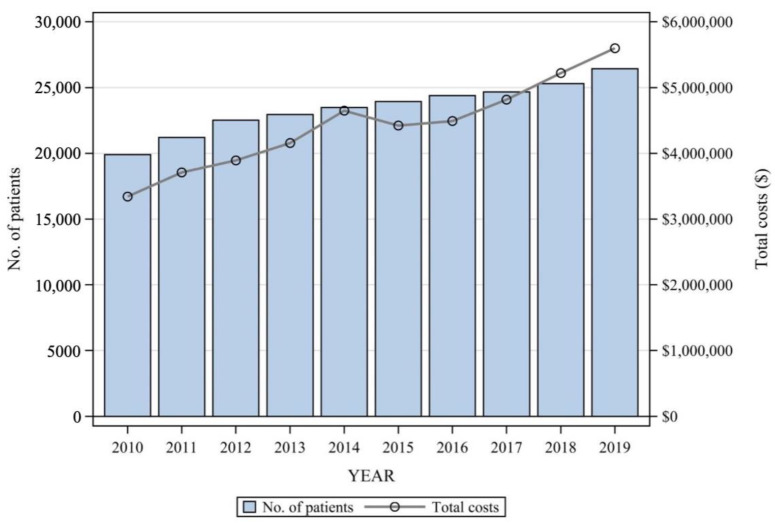
Medical cost of lumbar radiculopathy.

**Figure 3 healthcare-11-02353-f003:**
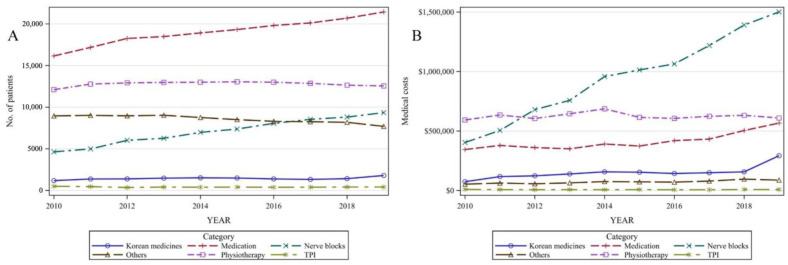
(**A**) Medical use of non-surgical treatment for lumbar radiculopathy. (**B**) Cost of non-surgical treatment for lumbar radiculopathy; TPI: trigger point injection.

**Figure 4 healthcare-11-02353-f004:**
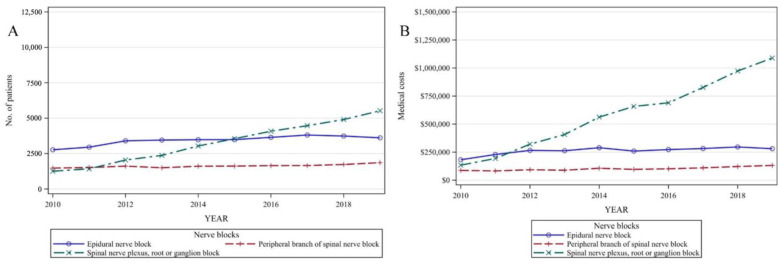
Medical use of nerve blocks. (**A**) Number of patients who received nerve blocks. (**B**) Total medical cost of nerve blocks.

**Table 1 healthcare-11-02353-t001:** Basic characteristics of patients.

Characteristics	No. of Patients	Percentage
**Total**	234,858	100.00
**Age**		
20–29	14,465	6.16
30–39	27,561	11.74
40–49	40,229	17.13
50–59	57,343	24.42
60–69	50,121	21.34
≥70	45,139	19.22
**Sex**		
Male	97,219	41.39
Female	137,639	58.61
**Payer type ***		
NHI	221,055	94.12
Medicaid	13,655	5.81
Others	148	0.06

* Others include military and veteran healthcare system. NHI: National Health Insurance.

**Table 2 healthcare-11-02353-t002:** Medication for lumbar radiculopathy.

Year	2010		2011		2012		2013		2014	
Category	No. of Patients (%)	Medical Costs	No. of Patients (%)	Medical Costs	No. of Patients (%)	Medical Costs	No. of Patients (%)	Medical Costs	No. of Patients (%)	Medical Costs
Opioids	6468 (32.49%)	USD 47,991	6796 (32.04%)	USD 58,008	7423 (32.96%)	USD 60,484	7573 (32.99%)	USD 62,775	7726 (32.88%)	USD 77,725
Non-opioid analgesics	14,057 (70.61%)	USD 237,719	14,814 (69.84%)	USD 260,964	15,699 (69.71%)	USD 235,957	15,889 (69.21%)	USD 236,299	16,099 (68.52%)	USD 268,784
Anesthetics	3976 (19.97%)	USD 3522	4407 (20.78%)	USD 4495	5352 (23.76%)	USD 4754	5724 (24.93%)	USD 4921	6254 (26.62%)	USD 5258
Gastrointestinal	12,616 (63.37%)	USD 124,552	13,429 (63.31%)	USD 136,958	14,390 (63.89%)	USD 132,835	14,685 (63.97%)	USD 140,187	15,007 (63.88%)	USD 159,086
Antipsychotic	2716 (13.64%)	USD 15,247	2664 (12.56%)	USD 18,991	2597 (11.53%)	USD 15,358	2472 (10.77%)	USD 19,303	2430 (10.34%)	USD 26,598
Antibiotics	789 (3.96%)	USD 13,786	787 (3.71%)	USD 12,240	786 (3.49%)	USD 10,874	837 (3.65%)	USD 11,469	864 (3.68%)	USD 12,538
Steroids	4425 (22.23%)	USD 6553	4963 (23.40%)	USD 7407	5718 (25.39%)	USD 7050	5715 (24.89%)	USD 5578	5700 (24.26%)	USD 5813
Others	8626 (43.33%)	USD 164,106	8873 (41.83%)	USD 174,297	9485 (42.11%)	USD 155,872	9847 (42.89%)	USD 164,890	10,362 (44.10%)	USD 184,074
**Year**	**2015**		**2016**		**2017**		**2018**		**2019**	
**Category**	**No. of Patients (%)**	**Medical Costs**	**No. of Patients (%)**	**Medical Costs**	**No. of Patients (%)**	**Medical Costs**	**No. of Patients (%)**	**Medical Costs**	**No. of Patients (%)**	**Medical Costs**
Opioids	7762 (32.42%)	USD 75,213	7967 (32.65%)	USD 75,318	7978 (32.33%)	USD 78,781	8244 (32.57%)	USD 94,459	8705 (32.92%)	USD 104,940
Non-opioid analgesics	16,399 (68.49%)	USD 247,407	16,872 (69.14%)	USD 267,065	17,037 (69.05%)	USD 289,043	17,560 (69.39%)	USD 323,767	18,322 (69.29%)	USD 377,972
Anesthetics	6689 (27.94%)	USD 5107	7367 (30.19%)	USD 5318	7880 (31.94%)	USD 5917	8309 (32.83%)	USD 6249	8911 (33.70%)	USD 6711
Gastrointestinal	15,262 (63.74%)	USD 149,845	15,652 (64.14%)	USD 158,084	15,973 (64.73%)	USD 168,681	16,329 (64.52%)	USD 186,856	17,090 (64.63%)	USD 201,417
Antipsychotics	2334 (9.75%)	USD 24,539	2062 (8.45%)	USD 28,057	1937 (7.85%)	USD 27,529	1907 (7.54%)	USD 28,564	1833 (6.93%)	USD 34,020
Antibiotics	849 (3.55%)	USD 10,516	827 (3.39%)	USD 13,511	739 (2.99%)	USD 11,259	777 (3.07%)	USD 10,016	681 (2.58%)	USD 8926
Steroids	5956 (24.88%)	USD 4977	6361 (26.07%)	USD 5536	6766 (27.42%)	USD 5465	6899 (27.26%)	USD 5875	7342 (27.77%)	USD 5280
Others	10,655 (44.50%)	USD 167,309	11,416 (46.78%)	USD 187,091	11,778 (47.73%)	USD 198,196	12,195 (48.19%)	USD 228,621	11,157 (42.19%)	USD 249,625

## Data Availability

Patient samples can be obtained via the website of HIRA by completing the End User Agreement of the Patient Samples. The patient samples are provided in a DVD (textfile) format, and a fee is charged for the samples. https://opendata.hira.or.kr/home.do (accessed on 17 July 2023).

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
