# Peer review of "Analysis of Conservative Treatment Trends for Lumbar Disc Herniation with Radiculopathy in Korea: A Population-Based Cross-Sectional Study"

_healthcare, 2023, doi:10.3390/healthcare11162353_

Round 1

Reviewer 1 Report

The manuscript has bare statistics and a statement of facts without real analysis.

There are no conclusions why this is happening and what needs to be done to improve the situation, namely, to improve the quality of treatment while reducing its cost.

Methods are not described, references are not informative and do not correspond to the original text.

For example, "bed rest, physiotherapy, nerve blocks, medications including opioids, non-opioid analgesics, psychotropic agents, acupuncture and manual therapy, and exercise;6,11,19"

In these articles there is not a word about physiotherapy, what methods and techniques are used. Without this, it is impossible to draw conclusions. For example, one cannot simply say that drugs are no longer in demand, there are thousands of different ones, it is important which drugs are no longer used and why.

Author Response

Reviewer 1

The manuscript has bare statistics and a statement of facts without real analysis.

There are no conclusions why this is happening and what needs to be done to improve the situation, namely, to improve the quality of treatment while reducing its cost.

Methods are not described, references are not informative and do not correspond to the original text.

For example, "bed rest, physiotherapy, nerve blocks, medications including opioids, non-opioid analgesics, psychotropic agents, acupuncture and manual therapy, and exercise;6,11,19" In these articles there is not a word about physiotherapy, what methods and techniques are used. Without this, it is impossible to draw conclusions. For example, one cannot simply say that drugs are no longer in demand, there are thousands of different ones, it is important which drugs are no longer used and why.

- We appreciate the reviewer’s comment. We agree that this paper is primarily built on statistics. The methods that this study employs are descriptive statistics, which is an important step in understanding the current status of healthcare utilization and medical decisions. We believe that this paper holds importance from the perspective that a shift of treatment choice is observed throughout the last decade, which needs to be reflected in the future studies building evidence and in healthcare policies.

Based on the reviewer’s comment, we revised the Discussion and Conclusion as follows:

Non-opioid analgesics were mainly used for medication. Interestingly, the proportion of patients who received nerve blocks showed a steadily increasing trend. Among the types of nerve blocks, the number of patients who received nerve block targeting the collection of nerves (root, plexus, and ganglion) increased significantly, outnumbering the use of epidural nerve blocks since 2015, and the associated costs exhibited a dramatic increase.

The result of this study indicates a change in nonsurgical treatment choices over the last decade towards specific interventions. While high rate of medication prescriptions and physiotherapy is similar to previous studies(9) on patients with lumbar IDD and radiculopathy, the evidence supporting medication is inconsistent despite the high utilization, possibly due to the nonspecific indications of these two treatments. This study showed an increase in the dramatic increase in the use of nerve blocks particularly the perineural approach to the spinal nerve plexus and dorsal root ganglion than the nerve block for radiculopathy.

The epidural injection is the most commonly used treatment for lumbar IDD globally. In the United States, the treatment is performed 9 million times annually.(22) A previous study in Korea in 2009 showed that epidural injection was the most commonly used treatment.(23) While this trend was partially observed in the earlier data of our study, this study also shows a shift in the preferred type of nerve blocks. Even after epidural injections, inflammation in the nerve root is thought to cause pain, which may have been the reason for this change in preference toward a more direct approach to the nerve root and the site of inflammation.(9, 24)

The continuous increase of anesthetics prescription along with steroids in this study may be related to the increase of nerve blocks. Interestingly, opioids which was one of the most frequently prescribed medication in patients with lumbar IDD and radiculopathy, was surpassed by anesthetics in 2018. The use of therapeutic agents in nerve blocks have been reported to include both anesthetics and steroids (23, 26), supporting the results reported in this study.

Previous studies on the effectiveness of epidural injection of steroids and/or anesthetics compared to conservative treatments such as oral medication show mixed results. Local injection was beneficial for some outcomes such as worse leg pain in short-term pain relief and intermediate-term follow-ups, but not during long-term follow-ups (27-30). On the other hand, lumbar plexus blocks have been shown to be more effective than epidural blocks in post-operative anesthesia in terms of surgeon’s satisfaction(31) and consumption of opiates (32). However, the lack of sufficient data on long-term efficacy and safety of lumbar plexus blocks compared to conservative treatment is evident. The increasing number of local injections and related health expenditures imply the need for reassessing clinical guidelines and health policies in line with the current evidence.

Socioeconomic status has been pointed out as a potential influence on the diagnosis, development, and pain outcome of lumbar IDD and radiculopathy in the previous studies (33-35). On the other hand, studies showed mixed results on the association between overall healthcare utilization and socioeconomic status (SES) (36, 37). This study investigated nonsurgical interventions which are covered by NHI, encompassing 98% of the Korean population, and did not include healthcare services not covered by NHI which may depend highly on SES. Future studies are suggested on a wider scope of management options of IDD and radiculopathy to examine potential differences in the utilization rate by socioeconomic groups.

This study holds significance as it is the first to analyze the current state and 10-year trends related to the use of non-surgical medical treatments for patients with lumbar IDD and radiculopathy based on claims data from the NHIS.

However, this study has some limitations. First, the study only included outpatients and did not account for treatment items not covered by the NHIS, which may limit the applicability of the findings to real-world clinical practice. Second, the NHIS claims data were segmented on an annual basis, which prevented the examination of patients' surgical history prior to the applicable year. Third, socioeconomic variable was not included in the database, providing limited information on the potential differences of treatments by socioeconomic status. Finally, patients were selected solely based on NHIS claims diagnosis codes, without considering the severity of lumbar IDD or self-reported pain intensity. Hence, it is crucial to exercise caution while interpreting and generalizing the study findings to broader populations. Further research is required on the comparative effectiveness of therapies such as lumbar plexus blocks to epidural injections and conservative treatments prior to making clinical decisions and in developing health policies.

  1. Conclusions

Over the 10-year study period, the number of patients who underwent nerve block treatments, particularly those who used nerve plexus, root and ganglion nerve blocks, steadily increased. Given the upward trend in the number of patients with lumbar IDD and radiculopathy and the corresponding healthcare costs, the study's findings call for a need to review current clinical practice guidelines to align them with the current evidence and formulating relevant healthcare policies in the future. Further research is required on the effectiveness of different types of local injections compared to conservative treatments.

Reviewer 2 Report

The authors have delivered a descriptive study on the conservative treatment for lumbar disc herniation, employing national health claim data from Korea. This research underscores the evolving trends in non-surgical treatment approaches for lumbar disc herniation, which will likely command significant interest from neurologists and orthopedists, particularly within Korea.

While the authors provide a comprehensive description of trends in conservative treatment for lumbar disc herniation in Korea, I find their conclusion misplaced. They suggest that their findings might "provide valuable evidence for revising clinical practice guidelines and formulating relevant healthcare policies." However, this is purely a descriptive study with no patient outcomes, such as pain relief or morbidity, accounted for. Consequently, it does not allow for determining an optimal treatment strategy. Even though the study reveals an increasing trend in conservative treatment, it doesn't necessarily imply that conservative treatment is superior to surgery. Additionally, identifying the best conservative treatment is not possible based on this study alone. I recommend the authors revise their conclusion to align it more accurately with their study's results.

Line 100 The term "Korean medicine" is ambiguous. Does it refer to traditional herbal medicine used in Korea? Please clarify this within the manuscript. If "Korean medicine" denotes Korean herbal medicine, it should be noted that "medication" as used in this manuscript does not encompass herbal medicine. This point should also be explicitly stated in the manuscript, as some Asian clinicians might assume that "medication" includes herbal medicine.

Line 140 The term "fona" is unclear to me. Based on the context, I assume it signifies "as a result." Please consider revising for clarity.

Line 256-262 This paragraph primarily summarizes the results, not a discussion. The authors' intended points for discussion in this paragraph remain unclear.

Supplementary file: The authors have classified medications using the Anatomical Therapeutic Chemical Classification (ATCC) code. Unfortunately, I suspect most clinicians are unfamiliar with the ATCC code. I suggest the authors provide the specific names of each medication alongside their respective ATCC codes.

Author Response

Reviewer 2

  1. The authors have delivered a descriptive study on the conservative treatment for lumbar disc herniation, employing national health claim data from Korea. This research underscores the evolving trends in non-surgical treatment approaches for lumbar disc herniation, which will likely command significant interest from neurologists and orthopedists, particularly within Korea.

- We thank the reviewer for the comment.

  1. While the authors provide a comprehensive description of trends in conservative treatment for lumbar disc herniation in Korea, I find their conclusion misplaced. They suggest that their findings might "provide valuable evidence for revising clinical practice guidelines and formulating relevant healthcare policies." However, this is purely a descriptive study with no patient outcomes, such as pain relief or morbidity, accounted for. Consequently, it does not allow for determining an optimal treatment strategy. Even though the study reveals an increasing trend in conservative treatment, it doesn't necessarily imply that conservative treatment is superior to surgery. Additionally, identifying the best conservative treatment is not possible based on this study alone. I recommend the authors revise their conclusion to align it more accurately with their study's results.

- We appreciate the reviewer’s comment. Based on the reviewer’s comment, we revised the Discussion and Conclusion as follows:

Non-opioid analgesics were mainly used for medication. Interestingly, the proportion of patients who received nerve blocks showed a steadily increasing trend. Among the types of nerve blocks, the number of patients who received nerve block targeting the collection of nerves (root, plexus, and ganglion) increased significantly, outnumbering the use of epidural nerve blocks since 2015, and the associated costs exhibited a dramatic increase.

The result of this study indicates a change in nonsurgical treatment choices over the last decade towards specific interventions. While high rate of medication prescriptions and physiotherapy is similar to previous studies(9) on patients with lumbar IDD and radiculopathy, the evidence supporting medication is inconsistent despite the high utilization, possibly due to the nonspecific indications of these two treatments. This study showed an increase in the dramatic increase in the use of nerve blocks particularly the perineural approach to the spinal nerve plexus and dorsal root ganglion than the nerve block for radiculopathy.

The epidural injection is the most commonly used treatment for lumbar IDD globally. In the United States, the treatment is performed 9 million times annually.(22) A previous study in Korea in 2009 showed that epidural injection was the most commonly used treatment.(23) While this trend was partially observed in the earlier data of our study, this study also shows a shift in the preferred type of nerve blocks. Even after epidural injections, inflammation in the nerve root is thought to cause pain, which may have been the reason for this change in preference toward a more direct approach to the nerve root and the site of inflammation.(9, 24)

The continuous increase of anesthetics prescription along with steroids in this study may be related to the increase of nerve blocks. Interestingly, opioids which was one of the most frequently prescribed medication in patients with lumbar IDD and radiculopathy, was surpassed by anesthetics in 2018. The use of therapeutic agents in nerve blocks have been reported to include both anesthetics and steroids (23, 26), supporting the results reported in this study.

Previous studies on the effectiveness of epidural injection of steroids and/or anesthetics compared to conservative treatments such as oral medication show mixed results. Local injection was beneficial for some outcomes such as worse leg pain in short-term pain relief and intermediate-term follow-ups, but not during long-term follow-ups (27-30). On the other hand, lumbar plexus blocks have been shown to be more effective than epidural blocks in post-operative anesthesia in terms of surgeon’s satisfaction(31) and consumption of opiates (32). However, the lack of sufficient data on long-term efficacy and safety of lumbar plexus blocks compared to conservative treatment is evident. The increasing number of local injections and related health expenditures imply the need for reassessing clinical guidelines and health policies in line with the current evidence.

Socioeconomic status has been pointed out as a potential influence on the diagnosis, development, and pain outcome of lumbar IDD and radiculopathy in the previous studies (33-35). On the other hand, studies showed mixed results on the association between overall healthcare utilization and socioeconomic status (SES) (36, 37). This study investigated nonsurgical interventions which are covered by NHI, encompassing 98% of the Korean population, and did not include healthcare services not covered by NHI which may depend highly on SES. Future studies are suggested on a wider scope of management options of IDD and radiculopathy to examine potential differences in the utilization rate by socioeconomic groups.

This study holds significance as it is the first to analyze the current state and 10-year trends related to the use of non-surgical medical treatments for patients with lumbar IDD and radiculopathy based on claims data from the NHIS.

However, this study has some limitations. First, the study only included outpatients and did not account for treatment items not covered by the NHIS, which may limit the applicability of the findings to real-world clinical practice. Second, the NHIS claims data were segmented on an annual basis, which prevented the examination of patients' surgical history prior to the applicable year. Third, socioeconomic variable was not included in the database, providing limited information on the potential differences of treatments by socioeconomic status. Finally, patients were selected solely based on NHIS claims diagnosis codes, without considering the severity of lumbar IDD or self-reported pain intensity. Hence, it is crucial to exercise caution while interpreting and generalizing the study findings to broader populations. Further research is required on the comparative effectiveness of therapies such as lumbar plexus blocks to epidural injections and conservative treatments prior to making clinical decisions and in developing health policies.

  1. Conclusions

Over the 10-year study period, the number of patients who underwent nerve block treatments, particularly those who used nerve plexus, root and ganglion nerve blocks, steadily increased. Given the upward trend in the number of patients with lumbar IDD and radiculopathy and the corresponding healthcare costs, the study's findings call for a need to review current clinical practice guidelines to align them with the current evidence and formulating relevant healthcare policies in the future. Further research is required on the effectiveness of different types of local injections compared to conservative treatments.

  1. Line 100 The term "Korean medicine" is ambiguous. Does it refer to traditional herbal medicine used in Korea? Please clarify this within the manuscript. If "Korean medicine" denotes Korean herbal medicine, it should be noted that "medication" as used in this manuscript does not encompass herbal medicine. This point should also be explicitly stated in the manuscript, as some Asian clinicians might assume that "medication" includes herbal medicine.

- We appreciate the reviewer’s comment. We revised two sections as follows:

Introduction:

Conservative (non-surgical) treatment includes a range of modalities, such as bed rest, physiotherapy, nerve blocks, medications including opioids, non-opioid analgesics, psychotropic agents, exercise, acupuncture and manual therapy, of which the latter two therapies are provided in Korean Medicine clinics in Korea.6,11,19

Study population:

The inclusion criteria were as follows: patients with one or more claims of lumbar IDD with radiculopathy (Korean Standard Classification of Diseases (KCD)-7 code: M511) as the primary diagnosis from 2010 to 2019, and those who visited medical institutions, including tertiary hospitals, general hospitals, hospitals, clinics, Korean Medicine (KM) hospitals or clinics which provide acupuncture, Tuina or Chuna (manual therapy), and herbal medicine. Only the adult patients aged ≥20 years with no missing data in the study variables were included in the study.

  1. Line 140 The term "fona" is unclear to me. Based on the context, I assume it signifies "as a result." Please consider revising for clarity.

- We thank the reviewer for the comment. We changed the sentence as follows:

Patients aged <20 years (n=4,156) were also excluded, resulting in 268,650 patients.

  1. Line 256-262 This paragraph primarily summarizes the results, not a discussion. The authors' intended points for discussion in this paragraph remain unclear.

- We agree with the reviewer that this paragraph belongs to the Results section than Discussion. Based on the reviewer’s comment, we revised the Results and Discussion as follows:

Results:

Among the types of prescribed medications, the use of anesthetics had the most significant increase over the 10-year period. In 2010, opioids accounted for the second-highest proportion of medications prescribed for therapeutic purposes in patients with lumbar IDD and radiculopathy. However, in 2018, patients who used anesthetics outnumbered those who used opioids.

Discussion:

The continuous increase of anesthetics prescription along with steroids in this study may be related to the increase of nerve blocks. Interestingly, opioids which was one of the most frequently prescribed medication in patients with lumbar IDD and radiculopathy, was surpassed by anesthetics in 2018. The use of therapeutic agents in nerve blocks have been reported to include both anesthetics and steroids (23, 26), supporting the results reported in this study.

  1. Supplementary file: The authors have classified medications using the Anatomical Therapeutic Chemical Classification (ATCC) code. Unfortunately, I suspect most clinicians are unfamiliar with the ATCC code. I suggest the authors provide the specific names of each medication alongside their respective ATCC codes.

- We appreciate the reviewer’s comment. We revised the table as the reviewer suggested:

Supplementary Table 1. Classification of medication

Category

Anatomical Therapeutic Chemical Classification code

Opioids

N02A (Opioids)

Non-opioid analgesics

M01A (Antiinflammatory and antirheumatic products, non-steroids), M01C (Specific antirheumatic agents), M02A (Topical products for joint and muscular pain), M03 (Muscle relaxants)

N02B (Other analgesics and antipyretics), N03A (Antiepileptics)

Aesthetics

N01 (Anesthetics)

Gastrointestinal

A02AA (Magnesium compounds), A02AA02 (Magnesium oxide), A02AA04 (Magnesium hydroxide), A02AB (Aluminium compounds), A02AB01 (Aluminium hydroxide), A02AB03 (Aluminium phosphate), A02AC01 (Calcium carbonate), A02AD01 (Ordinary salt combinations), A02AD03 (Almagate), A02AD04 (Hydrotalcite), A02AD05 (Almasilate), A02AX (Antacids, other combinations),

A02B (Drugs for peptic ulcer and gastro-oesophageal reflux disease (GORD)),

A02X (Other drugs for acid related disorders), A03 (Drugs for functional gastrointestinal disorders),

A06 (Drugs for constipation), A07 (Antidiarrheals, intestinal antiinflammatory/antiinfective agents), A09 (Digestives, incl. enzymes), A16 (Other alimentary tract and metabolism products)

Antipsychotics

N05 (Psycholeptics), N06 (Psychoanaleptics), N07 (Other nervous system drugs)

Antibiotics

D01 (Antifungals for dermatological use), D06A (Antibiotics for topical use), D06BB (Sulfonamides and other antibacterials for topical use), D06BX (Other topical antibiotics)

G01A (Antiinfectives and antiseptics for gynecological use)

J01 (Antibacterials for systemic use), J02 (Antimycotics for systemic use), J04 (Drugs for treatment of tuberculosis), J05 (Antivirals for systemic use)

Steroids

D07 (Corticosteroids for topical use), H02 (Corticosteroids for systemic use)

Others

A01A (Stomatological preparations), A05 (Bile and liver therapy), A10 (Drugs used in diabetes), A11 (Vitamins), A12 (Mineral supplements)

B01A (Antithrombotic agents), B02 (Antihemorrhagics), B03 (Antianemic preparations), B05 (Blood substitutes and perfusion solutions)

C (Cardiovascular system),

D02A (Emollients and protectives), D03A (Preparations for treatment of wounds and ulcers), D05 (Antipsoriatics), D06 (Antibiotics and chemotherapeutics for dermatological use), D08 (Antiseptics and disinfectants), D11 (Other dermatological preparations)

G02 (Other gynecologicals), G03 (Sex hormones and modulators of the genital system), G04 (Urologicals)

H01 (Pituitary and hypothalamic hormones and analogues), H03 (Thyroid therapy), H05 (Calcium homeostasis)

J06 (Immune sera and immunoglobulins)

L (Antineoplastic and immunomodulating agents), M04 (Antigout preparations), M05 (Drugs for treatment of bone diseases), M09 (Other drugs for disorders of the musculo-skeletal system), N02 (Analgesics), N04 (Antiparkinson drugs), P (Antiparasitic products, insecticides and repellents), R (Respiratory system), S (Sensory organs), V (Various)

Reviewer 3 Report

Thank you for your manuscript. It is always interesting to know the reality of other countries, although perhaps it would be necessary to provide more context for those who do not know the Korean reality and also to know if the data could be extrapolated to other countries.

For example, in Korea, most patients pay for their healthcare, whereas in Europe there are free or even universal healthcare systems. In addition, it is important to know the socioeconomic context of each sector of the Korean population, as it is possible that there are population groups that are less likely to seek medical services due to lack of resources. In this regard, we recommend adding two columns to Table 1: one indicating the percentage of the population in each group (with respect to the total Korean population) and another column indicating the socioeconomic level of each group.

In line 29 (Abstract) when "due to" is indicated, it is very important to note that concurrence does not imply causality. It may be that one treatment is used more than another because it has been indicated in the clinical guidelines, because it is in vogue among professionals, etc. There are multiple factors that should be studied.

In line 50-53, it is not clear to me that in all countries there is a desire to perform surgery after a 6-week failure of conservative treatment. It should be completed by indicating the time brackets.

On lines 68-73, could you justify why 2016-17 and 18 guides are used? Couldn't anything more current be used? In the same respect, you do an analysis of data from 2010 to 2019, could you explain why more current data is not used or why this study was not sent for publication in 2020?

As for the discussion and conclussions, it is solid, but needs improvement in some aspects:

1. Further explanation of clinical implications: Although it is mentioned that there is a trend toward more specific treatments rather than general options, it might be useful to explain the clinical implications of this trend. How might this affect treatment efficacy and patient satisfaction? Are there clear advantages or disadvantages to the use of more specific treatments?

2.Comparison with other studies: While briefly mentioning that some results are consistent with previous studies, it might be valuable to provide a brief comparison with other studies conducted in different countries or settings. Have similar trends been observed in other parts of the world? Are there significant differences between treatment patterns in Korea and other countries?

3. Discussion on the increase in the use of anesthetics: Given that a significant increase in the use of anesthetics compared to opioids was observed, it would be relevant to discuss possible reasons behind this trend. Are there any medical or safety reasons for this change? Have there been any changes in medical prescribing patterns that may have influenced this increase?

4. Consideration of limitations: While the limitations of the study are mentioned, it might be useful to discuss how these limitations might affect the interpretation of the results and the applicability of the findings in clinical practice. In addition, future research could be explored to address some of the identified limitations.

Author Response

Reviewer 3

  1. Thank you for your manuscript. It is always interesting to know the reality of other countries, although perhaps it would be necessary to provide more context for those who do not know the Korean reality and also to know if the data could be extrapolated to other countries.

- We thank the reviewer for the comment.

  1. For example, in Korea, most patients pay for their healthcare, whereas in Europe there are free or even universal healthcare systems. In addition, it is important to know the socioeconomic context of each sector of the Korean population, as it is possible that there are population groups that are less likely to seek medical services due to lack of resources. In this regard, we recommend adding two columns to Table 1: one indicating the percentage of the population in each group (with respect to the total Korean population) and another column indicating the socioeconomic level of each group.

- We appreciate the reviewer’s comment. We agree with the reviewer that an elaborate explanation on the healthcare system of Korea will help expand the readership. Korea has achieved universal healthcare systems (UHC) in the 1980s, including Korean Medicine which incorporates acupuncture, moxibustion, and manual therapy. The expenditure presented in this paper are all expenditures covered by national health insurance.

Furthermore, previous studies showed mixed results on the statistical association between utilization of Korean Medicine healthcare and socioeconomic status such as income, educational level, and type of insurance (National Health Insurance or Medical Aid). While this database does not include any socioeconomic variable, we understand the importance of the reviewer’s comment and revised the Methods and Discussion as follows:

Methods:

This study analyzed claims data from the Health Insurance Review and Assessment Service (HIRA) and HIRA-National Patient Sample (HIRA-NPS) data from January 2010 to December 2019. Due to universal health coverage in Korea, the National Health Insurance Service (NHIS) covers 98% of the national population in Korea.

The prescribed medications during inpatient and outpatient care were categorized according to the Anatomical Therapeutic Chemical classification system, and the results are presented in Supplementary Table 1. Healthcare expenditure covered by NHIS were analyzed by types of interventions.

Discussion:

Socioeconomic status has been pointed out as a potential influence on the diagnosis, development, and pain outcome of lumbar IDD and radiculopathy in the previous studies (27-29). On the other hand, studies showed mixed results on the association between overall healthcare utilization and socioeconomic status (SES) (30, 31). This study investigated nonsurgical interventions which are covered by NHI, encompassing 98% of the Korean population, and did not include healthcare services not covered by NHI which may depend highly on SES. Future studies are suggested on a wider scope of management options of IDD and radiculopathy to examine potential differences in the utilization rate by socioeconomic groups.

  1. In line 29 (Abstract) when "due to" is indicated, it is very important to note that concurrence does not imply causality. It may be that one treatment is used more than another because it has been indicated in the clinical guidelines, because it is in vogue among professionals, etc. There are multiple factors that should be studied.

- We appreciate the reviewer’s comment. Based on the reviewer’s comment, we revised the Abstract as follows:

Discussion:

In conclusion, the number of patients with radiculopathy who receive nerve blocks, particularly nerve plexus, root and ganglion nerve blocks, and related expenditure increased, implying a gradual shift of medical decision from general pain reduction to specific and targeted pain treatments.

  1. In line 50-53, it is not clear to me that in all countries there is a desire to perform surgery after a 6-week failure of conservative treatment. It should be completed by indicating the time brackets.
    - We appreciate the reviewer’s comment. Based on the reviewer’s comment, we revised the Introduction as follows:

Introduction:

According to clinical practice guidelines, surgery can only be considered as an option when there is no improvement in symptoms after conservative treatment for at least 6 weeks, if not upholding the option of continued nonsurgical treatments; furthermore, surgery should not be considered as an option for cases involving the development of neurological symptoms or cauda equina syndrome.

  1. On lines 68-73, could you justify why 2016-17 and 18 guides are used? Couldn't anything more current be used? In the same respect, you do an analysis of data from 2010 to 2019, could you explain why more current data is not used or why this study was not sent for publication in 2020?

- We appreciate the reviewer’s comment and understand his concern. The data of the year 2020 was released a few months ago according to the provider (Health Insurance Review and Assessment Service (HIRA)). This analysis was conducted in 2022 including the latest source data approved and available for research.

Regarding the clinical guidelines, we double-checked for updates of the guidelines we listed in the paper but only found that what is listed in the paper are the latest guidelines specifically developed for low back pain or intervertebral disc herniation (e.g., https://www.ihe.ca/research-programs/hta/aagap/lbp2). We will continue to search for newer guidelines in the meantime and make updates during further revisions and proofreading.

  1. As for the discussion and conclussions, it is solid, but needs improvement in some aspects:

6.1. Further explanation of clinical implications: Although it is mentioned that there is a trend toward more specific treatments rather than general options, it might be useful to explain the clinical implications of this trend. How might this affect treatment efficacy and patient satisfaction? Are there clear advantages or disadvantages to the use of more specific treatments?

- We appreciate the reviewer’s comment. Based on the reviewer’s comment, we revised the Discussion as follows:

Previous studies on the effectiveness of epidural injection of steroids and/or anes-thetics compared to conservative treatments such as oral medication show mixed results. Local injection was beneficial for some outcomes such as worse leg pain in short-term pain relief and intermediate-term follow-ups, but not during long-term follow-ups (27-30). On the other hand, lumbar plexus blocks have been shown to be more effective than epidural blocks in post-operative anesthesia in terms of surgeon’s satisfaction(31) and consumption of opiates (32). However, the lack of sufficient data on long-term efficacy and safety of lumbar plexus blocks compared to conservative treatment is evident. The increasing number of local injections and related health expenditures imply the need for reassessing clinical guidelines and health policies in line with the current evidence.

6.2.Comparison with other studies: While briefly mentioning that some results are consistent with previous studies, it might be valuable to provide a brief comparison with other studies conducted in different countries or settings. Have similar trends been observed in other parts of the world? Are there significant differences between treatment patterns in Korea and other countries?

- We appreciate the reviewer’s comment. Based on the reviewer’s comment, we revised the Discussion as follows:

The epidural injection is the most commonly used treatment for lumbar IDD glob-ally. In the United States, the treatment is performed 9 million times annually.(22) A previous study in Korea in 2009 showed that epidural injection was the most commonly used treatment.(23) While this trend was partially observed in the earlier data of our study, this study also shows a shift in the preferred type of nerve blocks.

6.3. Discussion on the increase in the use of anesthetics: Given that a significant increase in the use of anesthetics compared to opioids was observed, it would be relevant to discuss possible reasons behind this trend. Are there any medical or safety reasons for this change? Have there been any changes in medical prescribing patterns that may have influenced this increase?

- We appreciate the reviewer’s comment. Based on the reviewer’s comment, we revised the Discussion as follows:

Previous studies on the effectiveness of epidural injection of steroids and/or anes-thetics compared to conservative treatments such as oral medication show mixed results. Local injection was beneficial for some outcomes such as worse leg pain in short-term pain relief and intermediate-term follow-ups, but not during long-term follow-ups (27-30). On the other hand, lumbar plexus blocks have been shown to be more effective than epidural blocks in post-operative anesthesia in terms of surgeon’s satisfaction(31) and consumption of opiates (32). However, the lack of sufficient data on long-term efficacy and safety of lumbar plexus blocks compared to conservative treatment is evident. The increasing number of local injections and related health expenditures imply the need for reassessing clinical guidelines and health policies in line with the current evidence.

6.4. Consideration of limitations: While the limitations of the study are mentioned, it might be useful to discuss how these limitations might affect the interpretation of the results and the applicability of the findings in clinical practice. In addition, future research could be explored to address some of the identified limitations.

- We appreciate the reviewer’s comment. Based on the reviewer’s comment, we revised the Discussion and Conclusion as follows:

Further research is required on the comparative effectiveness of therapies such as lumbar plexus blocks to epidural injections and conservative treatments prior to making clinical decisions and in developing health policies.

Over the 10-year study period, the number of patients who underwent nerve block treatments, particularly those who used nerve plexus, root and ganglion nerve blocks, steadily increased. Given the upward trend in the number of patients with lumbar IDD and radiculopathy and the corresponding healthcare costs, the study's findings call for a need to review current clinical practice guidelines to align them with the current evidence and formulating relevant healthcare policies in the future. Further research is required on the effectiveness of different types of local injections compared to conservative treatments.

Round 2

Reviewer 1 Report

I reviewed the revised manuscript and my decision is to accept it for publication

Reviewer 2 Report

The manuscript is well-constructed, and all the issues I raised have been appropriately addressed. I believe the manuscript is now in a publishable form.

Reviewer 3 Report

We thank the authors for taking our recommendations into account. We believe that this has improved the manuscript and made it publishable.